# High-Speed Shift Register with Dual-Gated Thin-Film Transistors for a 31-Inch 4K AMOLED Display

**DOI:** 10.3390/mi13101696

**Published:** 2022-10-09

**Authors:** Rong Song, Yonghe Wu, Chengkai Lin, Kai Liu, Zhenjun Qing, Yingxiang Li, Yan Xue

**Affiliations:** 1School of Integrated Circuit, Shenzhen Polytechnic, Shenzhen 518055, China; 2Vocational Education College, South China Normal University, Guangzhou 510631, China; 3Provincial Key Laboratory of Informational Service for Rural Area of Southwestern Hunan, Shaoyang University, Shaoyang 422004, China

**Keywords:** shift register, dual-gated thin film transistor, rising time, 31-inch 4K AMOLED display

## Abstract

In this work, a promising dual-gated thin film transistor (TFT) structure has been proposed and introduced in the shift register (SR)-integrated circuits to reduce the rising time. The threshold voltage can be simultaneously changed by the top gate and the bottom gate in the proposed dual-gated TFTs. When the SR circuits start to export the scan signals in the displays, the driving currents in the SR circuits are increased by switching the working station of driving TFTs from the enhancement characterization to the depletion characterization. Subsequently, the detailed smart spice simulation has been used to study the function of the proposed SR circuits. In the next step, the proposed SR circuits have been fabricated in a G4.5 active-matrix organic light-emitting diode manufacture factory. The simulated and experimental results indicate that the shift register pulses with the full swing amplitude can be obtained in the SR circuits. Moreover, in contrast to the conventional SR circuits employing with the single-gated TFTs, it has been found that the rising time of the output signals can be reduced from 3.75 μs to 1.23 μs in the proposed SR circuits with the dual-gated TFTs, thus exhibiting the significant improvement of the driving force in the proposed SR circuits. Finally, we demonstrated a 31-inch 4K AMOLED display with the proposed SR circuits.

## 1. Introduction

The flat panel displays high-resolution images, and these are used as an exchange platform of information, and they have been widely developed in various fields, including automotives, flexible electronics, smartphones, and televisions [1,2,3,4]. Up to now, the mainstream television resolution is 2K, while 4K is the next generation mainstream television technology [5,6]. However, when the resolution increases from 2K to 4K, the resistance and capacitance loading (RC loading) in the scan lines of the displays will increase significantly [7,8,9,10]. Therefore, it is a challenge to design these displays with a 4K resolution because the rising time is rather long.

Generally, two well-developed techniques, such as the reduction in the RC loading and the improvement of the driving force, have been used to reduce the rising time in the displays. To date, tremendous efforts have been made to decrease the RC loading in the displays. For example, the researchers at Beijing University have developed a top-gate TFT structure instead of a traditional back-channel etch TFT structure [11,12]. The non-overlap between the gate and the source electrode causes the reduction in the parasitic capacitance in the top-gated TFTs [13]. The introduction of the copper process instead of the aluminum process is a promising method to reduce the RC loading in the displays [14]. However, it is still a challenge for IC researchers to design 4K displays because of the long rising time. The improvement of the driving force in the displays is another method that is used to reduce the rising time.

In recent years, the direct fabrication of shift register (SR) circuits using TFT-integrated technology instead of the traditional driver ICs has attracted much attention from researchers worldwide [15,16,17,18]. For example, the researchers from Kyung Hee University have developed a simple SR circuit with a long lifetime. In the circuit, the lifetime can be increased from 1.7 years to 17.5 years [19]. C. W. Liao et al. have proposed a simple buffer structure in their SR circuit to suppress the feedthrough effects from the clock signals [20]. In order to improve the reliability of it, Z. J. Hu et al. have designed an amorphous silicon SR circuit with a threshold voltage shift compensable low-level holding unit [21]. M. Mativenga et al. have prepared a simple SR circuit on a plastic substrate [22]. In this circuit, the lifetime of the SR circuit can be improved to ten years. However, the introduction of the SR circuits in the displays should not only be simple, but also practical. When the display starts to work, the rising time from the output signals in the SR circuits should be as short as possible. To increase the driving force in the scan signals, it is essential to increase the TFT size in the SR circuits. However, it is difficult for the display to obtain a narrow border because the TFTs would occupy much of the layout area [23]. It should be mentioned that the use of a high driving voltage can increase the driving force in the SR circuits. However, the high voltage would result in a high level of power consumption [24]. It is desirable to develop other technologies to improve the driving force in the SR circuits.

Recently, due to their high mobility, dual-gated TFTs have been widely used in the integrated circuits [25,26,27,28]. For example, in our previous study, a dual-gated TFT structure that was based on the active-matrix organic light-emitting diode (AMOLED) manufacturing process has been introduced [29]. The threshold voltage (Vth) can be controlled by changing the voltage at the bottom gate. Meanwhile, the light stability of the TFTs can be improved because both the bottom and top gate can prevent the irradiation of light that comes from outside to the active layer. In this work, the dual-gated TFTs have been used to reduce the rising time in the SR circuits. Subsequently, the 48-stage SR circuits have been fabricated in a G4.5 AMOLED manufacture factory, and the circuit function can be tested by a jig.

## 2. TFT Performance

The cross-sectional schematic diagram of the dual-gated IGZO-TFTs is shown in Figure 1. The TFTs are fabricated through a commercial white AMOLED process. In the driving TFTs, the M0 layer not only blocks the light emitting from the bottom to the indium–gallium–zinc oxide (IGZO) semiconductor layer, but it act as a second gate electrode. Both the top gate and the bottom gate are connected to the internal nodes in the SR circuits. It should be noted that the parasitic capacitances in the dual-gated TFTs are much higher than those in the conventional single-gated TFTs. Therefore, the other TFTs adopt the conventional single-gated structure to avoid increasing the dynamic power consumption in the SR circuits.

The TFT process can be described as follows: The M0 is firstly deposited by sputtering, and it is patterned by the wet etching process on a bare glass substrate. Then, a 200 nm-thick buffer (SiOx) layer is deposited on the M0 layer with the deposition temperature of 300 °C. A 50 nm-thick IGZO layer (In:Ga:Zn = 1:1:1) is deposited on the buffer layer by the reactive sputtering at 200 °C, and it is patterned by the wet etching process. The detailed reactive sputtering process can be seen elsewhere [30]. Subsequently, the 200 nm-thick gate insulator (GI) layer and the 600 nm-thick M1 electrode layer are deposited on the IGZO layer. After the wet etching if performed, a 200 nm-thick SiOx interlayer dielectrics (ILD) layer is deposited and patterned on the surface of the M1 one. In the next step, the M2 layer is prepared to form the S/D electrodes. After the deposition of 300 nm-thick SiOx passivation (PV) layer, the TFTs are annealed under 250 °C for 1 h.

To study the performance of the dual-gated TFTs, the Keithley-4200 is used to test the I-V performance. The channel width and length are 20 μm and 8 μm, respectively. The TFTs are annealed under 60 °C and a N_2_ atmosphere before the I-V test is performed. The influence of the M0 bias voltage on the Vth of dual-gate TFTs is shown in Figure 2. The voltage at the top gate and source electrode was set to 0. When the voltage in the M0 (VLS) one changed from −15 V to 15 V, Vth shifted linearly from 8.5 V to −4 V, and the working mode of TFTs changed from the enhancement characterization to the depletion characterization.

## 3. SR Circuit

Figure 3a shows the diagram of a basic loop unit in the proposed SR circuits. The whole circuits contain 12 basic units. A pulse signal was used to start the SR circuits at the first stage. The clock signals (CK1~CK4), the duty ratios of which are 20% (Figure 3b), were used to pull up the output signals. The VGH and VGL were set to a high-level voltage and a low-level voltage, respectively.

Figure 4 exhibits the schematic and the time diagrams of the SR circuit. It is well known that IGZO-TFTs often operate as depletion-mode devices [31]. Moreover, the Vth shifts toward the negative direction continuously under an electrical and illumination stress [32]. In our previous work, a stable SR circuit has been proposed, and it was used in a 31-inch 4K flexible display [7]. However, the output voltage was reduced from 19 V to 17 V when the operation time increased to 500 h. In order to reduce the rising time, we introduce the dual-gated TFT structure in the SR circuits.

In this circuit, a feedback unit (T6) was used to reduce the leakage current that was caused by the IGZO-TFTs, and a pull-up control unit (T11, T12) was used to charge the storage capacitance (Cb). A pull-up unit was used to charge the scan signal G(n) and cascade signal M(n). It should be pointed out that the G(n) is the output signal from the SR circuit, and it acts as the scan signal in AMOLED displays. The function of M(n) not only acts as the step-in signal from the next SR circuit, but the feedback signal for the previous SR circuit. A pull-down unit (T31~T33) was used to release the high voltage in Q and G(n). The use of a pull-down holding unit (T41~T45) was to maintain the low voltage in Q and G(n). A reverse unit (T51~T54) was used to control the voltage of the gate electrode in the pull-down unit. The operation can be described as follows.

The first step is (P1 stage): The voltage of M(n − 1) is VGH, and the voltage in Q (V_Q_) is charged to VGH–Vth1–Vth2. Vth1 and Vth2 are the threshold voltage of T11 and T12, respectively. Due to the low voltage in V_V2_, T21~T23 are turned on to keep V_M(n)_ and V_G(n)_ at VGL. T52 and T54 are turned on, and the voltage in OB is released to a low voltage. The second step is (P2): V_V1_ is changed to VGL. T11 and T12 are turned off. At this time, V_Q_ maintains VGH–Vth1–Vth2. The third step is (P3): V_V2_ is changed to VGH. As a result, V_Q_ is coupled to a higher voltage and V_G(n)_ is charged to VGH. Meanwhile, V_M(n)_ is transferred to VGH, and the Vth shifts toward the negative direction in T21, as shown in Figure 2. Therefore, the driving current in T21 can be improved. The fourth step is: V2 is changed to VGL. As a result, G(n) is discharged to VGL through T21. The fifth step is: M(n + 1) is changed to VGH, T31~T33 are turned on and V_Q_ is discharged to VGL. Table 1 presents the detailed parameters of the SR circuits.

The Smart Spice simulator was used to make the accurate SPICE model. The initial model parameters were mainly determined by the measured TFT characteristics. Figure 5a shows the transfer curves of the TFT with the device size of W/L = 20 μm/8 μm. The model process can be described as follows: Firstly, we extracted the RPI model with the parameter from the measured oxide TFT characteristics. Secondly, the logical ternary condition was used to eliminate the noise-like characteristics. Finally, the deviation between the measured TFT characteristics and the simulated transfer curves were set to <5%. It can be seen that the simulated transfer curves of the IGZO-TFTs fit well with the experimental results of the fabricated TFTs in Figure 5a. The Smart Spice (Silvaco) simulation was used to testify the feasibility of the SR circuits. As shown in Figure 5b, the period in the CKs was set to 60 μs and the waveform of the output signals can be transformed in a step-by-step process with the pulse width of 12 μs. Figure 6 indicates the simulated output results. When the Vth increases from 0.5 V to 5.5 V, the distortion of the output signal has taken place for the SR circuit without the dual-gated TFT. On the contrary, the pulse signal in the proposed SR circuit is almost unchanged, indicating that the dual-gated TFT can successfully improve the driving force.

## 4. Results and Discussion

The function has been verified through fabricating the proposed SR circuits in a 4.5-generation AMOLED manufacture factory. The production of the whole SR circuits included 48 stages, and the optical microscope of them can be seen in Figure 7. The height and width of each SR circuit are 3500 μm and 280 μm, respectively. It should be mentioned that there are several test pads connecting to the output signals (G1, G10, G20, G30, G46) which are generated by the SR circuits.

Figure 8 shows the output pulses of the SR circuits without (Figure 8a) and with the dual-gated TFTs (Figure 8b). In these figures, the green line represents the waveform of the internal Q node (Figure 4a). The blue line is the output waveform of the 46th SR circuit (Figure 4b). The SR circuits have been used in the 31-inch 4K flexible displays with the conventional single-gated TFTs [7]. However, it has been found that the rising time increased monotonously when the SR circuits continued to work. Moreover, the SR circuits can hardly be used in 8K displays because of the long rising time. Figure 8a shows the output signals of the SR circuits with the conventional single-gated TFTs. It can be seen that the initial rising time (RT) is 3.75 μs and 1.23 μs for the 46th SR circuit (G46), respectively. It is well known that the charge time for the pixel circuit is reduced from 7.4 μs to 3.7 μs when the resolution improves from 4K to 8K. Obviously, the RT in the conventional SR circuits is longer than the charge time of the 8K displays. Therefore, the dual-gated TFT technology has been developed to reduce the rising time (Figure 4a). In the proposed SR circuits, when the M(n) was changed to VGH (20 V), the threshold voltage of driving TFT (T21) reduced to −7.5 V (Figure 2). Meanwhile, the driving current is largely dependent on the threshold voltage. Therefore, the rising time can be improved. In the proposed SR circuits, the experimental waveforms of the internal Q node and output pulse (the 46th SR circuit) can be seen in Figure 8b. In contrast to the output waveforms of the SR circuit without the dual-gate TFTs (Figure 8a), the faster rising time in the proposed SR circuits (1.23 μm) confirms that the driving force in the proposed circuit is much stronger. It can be seen that the pulses are smooth without any distortion, indicating the application potentiality of the dual-gated TFTs in displays.

The SR circuits with the dual-gated TFTs can improve the charging speed of the AMOLED displays. Figure 9a shows the schematic diagram of the pixel circuit, which have been used in our AMOLED displays [7]. The G(n) is the output signal from the SR circuits. The data voltage are generated by the external data ICs. It is well known that the image quality is largely dependent on the charging rate. Therefore, the data voltage must transfer to the internal node P, completely. As shown in Figure 9b, the charging voltage in the pixel is reduced when the rising time increases from the black line to the red dotted line. Obviously, it is necessary to reduce the rising time of G(n).

After the function test, the proposed SR circuits were fabricated in a 31-inch AMOLED display. The display structure can be seen in Figure 10a. Figure 10b shows a photograph of the display image of the 31-inch AMOLED display, which was fabricated in the 4.5-generation AMOLED display manufacture factory. The image is smooth and clear, indicating the high quality of the proposed SR circuits. The specifications of the display can be seen in Table 2.

## 5. Conclusions

In this paper, a promising dual-gated TFT structure, which was fabricated in a commercialized IGZO-TFT manufacture factory, has been proposed and designed. In this dual-gated TFT structure, the threshold voltage can be varied by changing the voltage at the bottom gate. To reduce the rising time, the replacement of the dual-gated structure to the conventional single-gated structure in the driving TFTs allows the improvement of the driving force in the SR circuits. Each SR circuit, including eighteen TFTs and one capacitance, has been utilized to testify the function of the dual-gated TFTs. When the SR circuits start to export the pulse signals, the working mode of the driving TFTs changes from the enhancement characterization to the depletion characterization, and the driving current can be increased. The experimental results indicate that the RT has been reduced from 3.75 μs to 1.23 μs by introducing the dual-gated TFTs. Moreover, a 31-inch 4K AMOLED display can be successfully driven by the proposed SR circuits, demonstrating the application potentiality of the dual-gate TFTs in AMOLED displays with a large size and high resolution.

## Figures and Tables

**Figure 1 micromachines-13-01696-f001:**
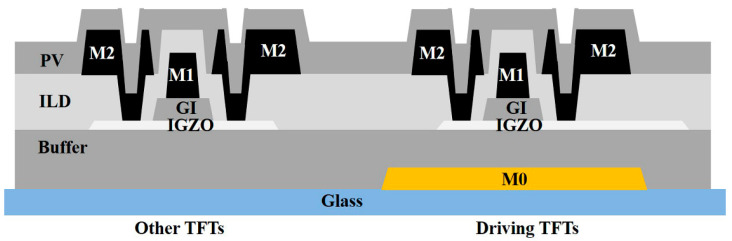
The cross-sectional diagram of dual-gated TFT.

**Figure 2 micromachines-13-01696-f002:**
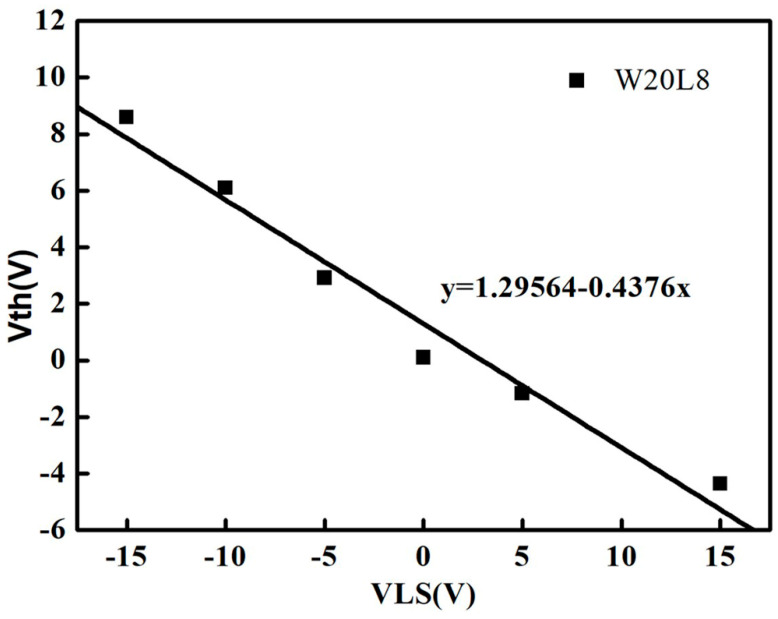
The effect of M0 voltage to Vth in dual-gate TFTs.

**Figure 3 micromachines-13-01696-f003:**
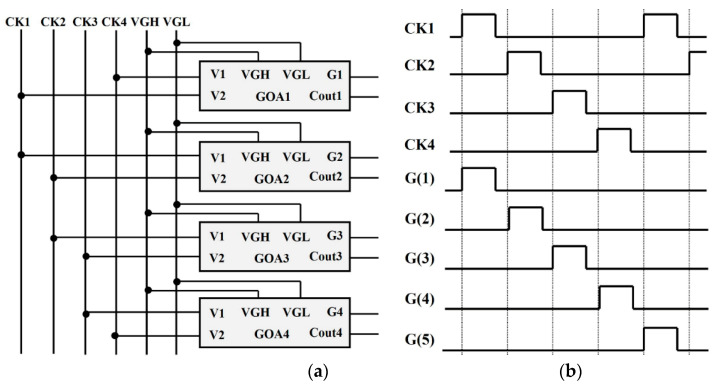
(**a**) Block diagram of a basic loop unit in SR circuit; (**b**) The diagram of clock signals.

**Figure 4 micromachines-13-01696-f004:**
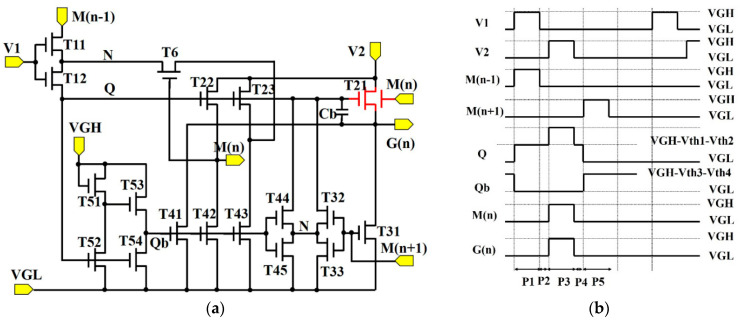
(**a**) The schematic diagram and (**b**) operation waveform of the proposed shift register circuit.

**Figure 5 micromachines-13-01696-f005:**
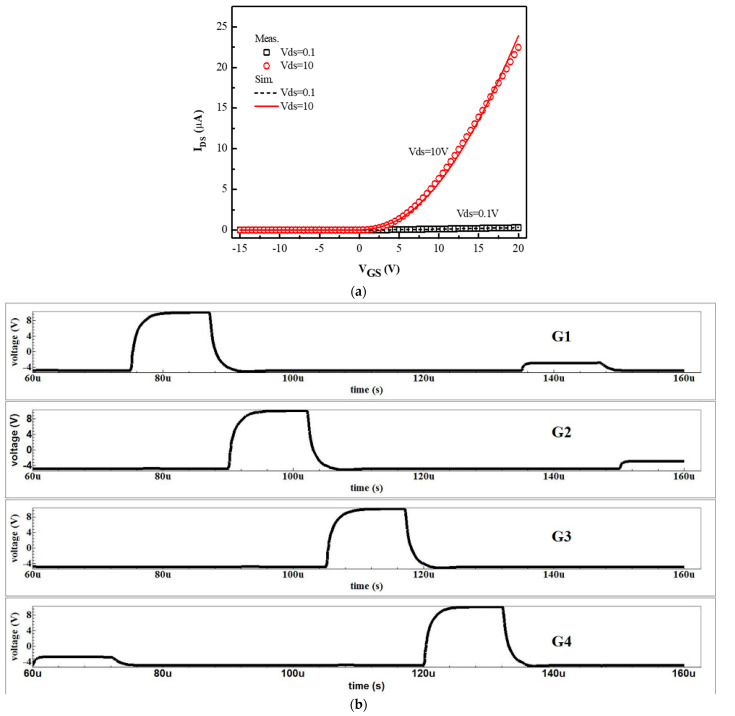
(**a**) The experimental result and device model of the transfer characteristics of IGZO-TFTs; (**b**) the output waveforms of the proposed SR circuits.

**Figure 6 micromachines-13-01696-f006:**
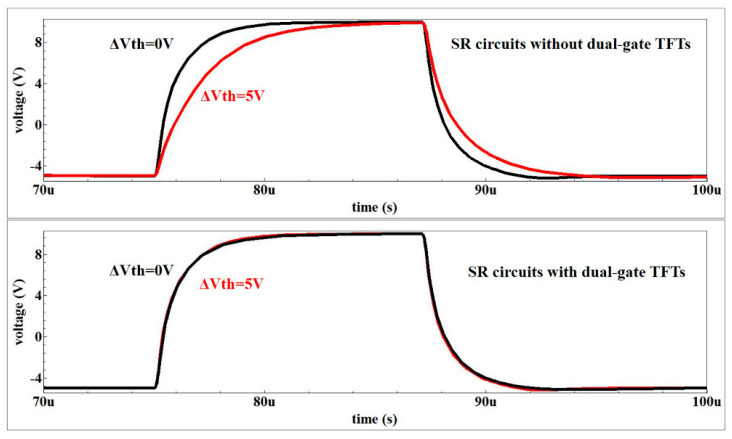
The effect of M0 voltage to Vth in dual-gate TFTs.

**Figure 7 micromachines-13-01696-f007:**
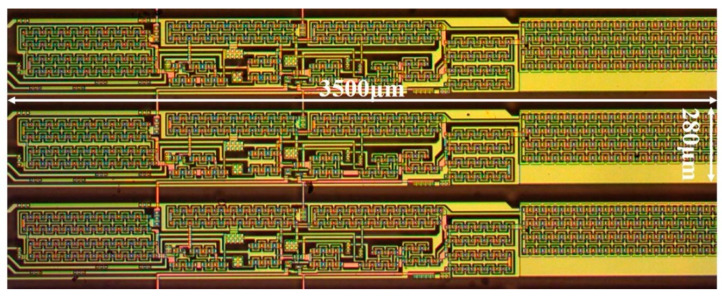
The optical microscope of the proposed SR circuits.

**Figure 8 micromachines-13-01696-f008:**
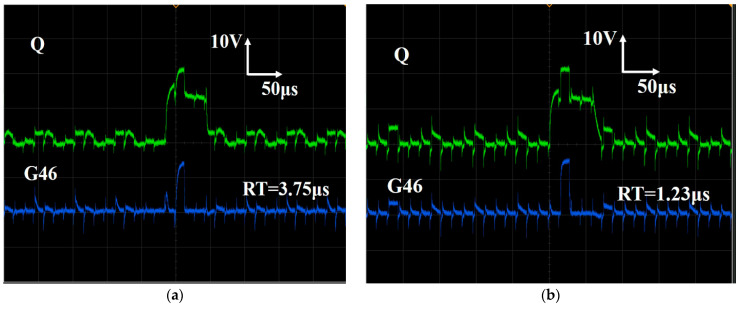
(**a**) The output signal generated by the SR circuit without dual-gate TFTs; (**b**) the output signal generated by the SR circuit with dual-gate TFTs.

**Figure 9 micromachines-13-01696-f009:**
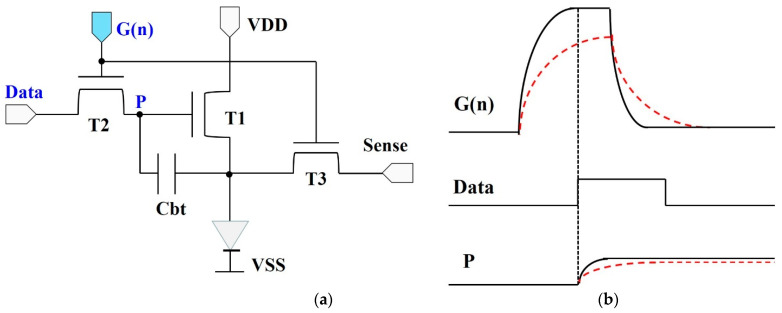
(**a**) The schematic diagram and (**b**) the charging diagram of pixel circuit in AMOLED displays.

**Figure 10 micromachines-13-01696-f010:**
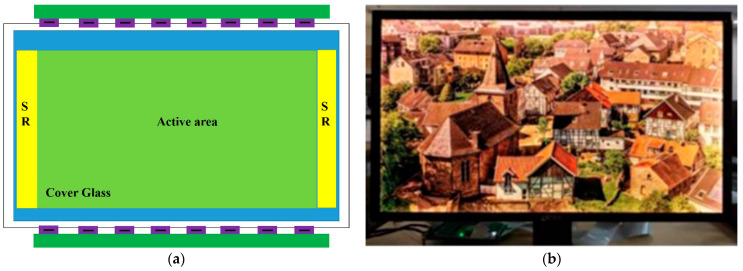
(**a**) Structure of AMOLED display; (**b**) Image of 31-inch 4K AMOLED display.

**Table 1 micromachines-13-01696-t001:** The parameters of the proposed SR circuit.

Design Parameter	Value
CKs	−5~10 V
VGH	10 V
VGL	−5 V
T11/T12/T22/T23	500 μm/8 μm
T21	2500 μm/8 μm
T31/T32/T44/T45	150 μm/8 μm
T33/T41/T42/T43	300 μm/8 μm
T51/T6	20 μm/8 μm
T52/T53/T54	80 μm/8 μm

**Table 2 micromachines-13-01696-t002:** Details of the proposed 31-inch AMOLED display.

Design Parameter	Value
Display size	31-inch
Border Width	7.5 mm
Frame Rate	60 Hz
Resolution	4K (3840 × 2160)
TFT Type	Dual-gate and Top-gate
OLED Process	Ink jet printer
Driver	SR circuit

## Data Availability

All data generated or analyzed during this study are included in this article.

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
