# Peer review of "High-Speed Shift Register with Dual-Gated Thin-Film Transistors for a 31-Inch 4K AMOLED Display"

_micromachines, 2022, doi:10.3390/mi13101696_

Round 1

Reviewer 1 Report

  In this manuscript, a shift register consisted of dual-gate TFTs is investigated to improve the response speed of the corresponding AMOLED panels. First of all, the English expression in this manuscript is so poor that audiences are hard to understand it; grammar and format mistakes are here and there. So, a complete proofreading by an English expert sounds like necessary. Besides, the technical content also needs major revision.

1.     The references about recent advances in GOA technology must be added in the revised manuscript.

2.     Did the authors propose the circuit shown in Fig. 4, or just make some changes on a conventional GOA circuit? This point must be clearly addressed in the revised manuscript?

3.     Are the curves shown in Fig. 5 the simulation results? If so, please provide the related simulation details (device models, simulation tools…).

4.     Please include more details about Fig. 8. For instance, what do the green lines and blue lines mean? What are the differences between Fig. 8(a) and Fig. 8(b)?

5.     A qualitative explain must be given for why the dual-gate-TFT-circuit improved the response speed of AMOLED panels.

Author Response

Revisions in response to the Editors’ comments:

  1. Reviewer1 wrote:

In this manuscript, a shift register consisted of dual-gate TFTs is investigated to improve the response speed of the corresponding AMOLED panels. First of all, the English expression in this manuscript is so poor that audiences are hard to understand it; grammar and format mistakes are here and there. So, a complete proofreading by an English expert sounds like necessary.

Our Response:

The reviewer’s comment is very important for our work. We totally agree with the suggestion and

improve the English. The changes have been highlighted in yellow:

Corresponding change in manuscript: Yes

Location of Change: Page 1-Page 9

  1. Reviewer1 wrote:

The references about recent advances in GOA technology must be added in the revised manuscript.

Our Response:

We greatly appreciate this question, which help us further improve paper quality. We totally agree with the suggestion and increase the recent advances in SR technology as follow:

For example, the researchers from Kyung Hee university have developed a simple SR circuit with a long lifetime. In the circuit, the lifetime can be increased from 1.7 year to 17.5 year [19]. C. W. Liao et al. have proposed a simple buffer structure in their SR circuit to suppress the feedthrough effects from the clock signals [20]. In order to improve the reliability, Z. J. Hu et al. have designed an amorphous silicon SR circuit with the threshold voltage shift compensable low-level holding unit [21]. M. Mativenga et al. have prepared a simple SR circuit on the plastic substrate [22]. In this circuit, the lifetime of the SR circuit can be improved to ten years. However, the introduction of the SR circuits in displays should not only be simple but also practical. When the display starts to work, the rising time from the output signals in the SR circuits should be as short as possible.

Location of Change: Page 2    line 47-56

  1. Review1 wrote:

Did the authors propose the circuit shown in Fig. 4, or just make some changes on a conventional GOA circuit? This point must be clearly addressed in the revised manuscript?

Our Response:

We greatly appreciate this question, which help us further improve paper quality. We totally agree with the suggestion and wrote that in Page 4:

In our previous work, a stable SR circuit has been proposed and used in a 31-inch 4K flexible display [7]. However, the output voltage is reduced from 19V to 17V when the operation time increases to 500h. In order to reduce the rising time, we introduce the dual-gated TFT structure in the SR circuits.

Location of Change: Page 4   line 115-117

  1. Reviewer1 wrote:

Are the curves shown in Fig. 5 the simulation results? If so, please provide the related simulation details (device models, simulation tools…).

Our Response:

We greatly appreciate this question, which help us further improve paper quality. We totally agree with the suggestion and wrote that in page 5:

Smart Spice (Silvaco) simulation is used to testify the feasibility of the SR circuits. As shown in Fig. 5a, the simulated transfer curves of IGZO-TFTs fit well with the experimental results of the fabricated TFTs (w=20mm, L=8mm)

Location of Change: Page5  line 143-145

(a)

Figure 5. (a) The experimental result and device model of the transfer characteristics of IGZO-TFTs

  1. Reviewer1 wrote:

Please include more details about Fig. 8. For instance, what do the green lines and blue lines mean? What are the differences between Fig. 8(a) and Fig. 8(b)?

Our Response:

We greatly appreciate this question, which help us further improve paper quality. We totally agree with the suggestion and wrote that:

Fig.8 shows the output pulses of the SR circuits without (Fig.8a) and with the dual-gated TFTs (Fig.8b). In these figures, the green line represents the waveform of the internal Q node (Fig.4a). The blue line is the output waveform of the 46th SR circuit (Fig.4b). The SR circuits have been used in the 31-inch 4K flexible displays with the conventional single-gated TFTs [7]. However, it has been found out that the rising time increased monotonously when the SR circuits continued to work. Moreover, the SR circuits can hardly be used in 8K displays because of the long rising time. Fig.8a shows the output signals of the SR circuits with the conventional single-gated TFTs. It can be seen that the initial rising time (RT) is 3.75μs and 1.23μs for the 46th SR circuit (G46), respectively. It is well known that the charge time for the pixel circuit is reduced from 7.4ms to 3.7ms when the resolution improves from 4K to 8K. Obviously, the RT in the conventional SR circuits is longer than the charge time of 8K displays. Therefore, the dual-gated TFT technology has been developed to reduce the rising time (Fig.4a). In the proposed SR circuits, when the M(n) is changed to VGH (20V), the threshold voltage of driving TFT (T21) is reduced to -7.5V (Fig.2). Meanwhile, the driving current is largely dependent on the threshold voltage. Therefore, the rising time can be improved. In the proposed SR circuits, the experimental waveforms of the internal Q node and output pulse (the 46th SR circuit) can be seen in Fig.8b. In contrast to the output waveforms of the SR circuit without dual-gate TFTs (Fig.8a), the faster rising time in the proposed SR circuits (1.23mm) confirms that the driving force in the proposed circuit is much stronger. It can be seen that the pulses are smooth without any distortion, indicating the application potentiality of the dual-gated TFTs in displays.

Location of Change: Page7, Line 167-185

  1. Reviewer1 wrote:

 A qualitative explain must be given for why the dual-gate-TFT-circuit improved the response speed of AMOLED panels.

Our Response:

We greatly appreciate this question, which help us further improve paper quality. We totally agree with the suggestion and wrote that in page 9:

In our previous work, a stable SR circuit has been proposed and used in a 31-inch 4K flexible display [7]. However, the output voltage is reduced from 19V to 17V when the operation time increases to 500h. In order to reduce the rising time, we introduce the dual-gated TFT structure in the SR circuits.

Location of Change: Page 4, Line 115-117

The dual-gated TFT circuit can improve the charging speed of display. Fig.9a shows the schematic diagram of the pixel circuit, which have been widely used in our AMOLED displays [7]. The G(n) is the output signal from the SR circuits. The data voltage is generated by the external data ICs. It is well known that the image quality is largely dependent on the charging rate. Therefore, the data voltage must transfer to the internal node P completely. As shown in Fig.9b, the charging voltage in pixel is reduced when the rising time increases from the black line to the red dotted line. Obviously, it is necessary to reduce the rising time of G(n).

After the function test, the proposed SR circuits were fabricated in a 31-inch AMOLED display

Location of Change: Page 9, Line 193-200

(a)

(b)

Figure 9. (a) The schematic diagram and (b) the charging diagram of pixel circuit in AMOLED displays

Reviewer 2 Report

The following comments are suggested for authors of the manuscript No. micromachines-1943271.

After reviewing your assigned manuscript entitled with High-Speed Shift Register with dual-gate thin-film transistors 2 for 31-inch 4K AMOLED display, I will give the following questions to the authors for revisions:

 1. 2. TFT performance, lines 85-86. When the voltage in M0 changes from -15V to 15V, Vth shifts linearly from 8.5V to -2.5V, ….. Please check Vth shifts linearly from 8.5V to -4 V in Fig.2.

2. Line 90. What’s the meaning of VLS of the x-axis lable in figure 2. Please define the VLS in the manuscript.

3. Lines 104-105. Please define the meanings of the cascade signal M(n) and Q point.

5. Is it the same working point of Qb in Figs. 4 (a) with the working point of QB in Fig. 4(b).

6. Lines 113-115. The second step (P2): VV1 is changed to VGL. T11 and T12 are turned off. At this time, VQ maintains VGH-VGL1-VGL2. Please check if VQ maintains VGH-Vth1-Vth2.

7. Please revise 60 HZ to 60 Hz in Table2.

Author Response

  1. Reviewer2 wrote:

TFT performance, lines 85-86. When the voltage in M0 changes from -15V to 15V, Vth shifts linearly from 8.5V to -2.5V, ….. Please check Vth shifts linearly from 8.5V to -4 V in Fig.2.

Our Response:

We greatly appreciate this question, which help us further improve paper quality. We totally agree with the suggestion and wrote that in page 4-6:

When the voltage in M0 (VLS) changes from -15V to 15V, Vth shifts linearly from 8.5V to -4V, and the working mode of TFTs changes from the enhancement characterization to the depletion characterization.

Location of Change: Page 2-3, Line 97-99

  1. Reviewer2 wrote:

What’s the meaning of VLS of the x-axis lable in figure 2. Please define the VLS in the manuscript.

Our Response:

We greatly appreciate this question, which help us further improve paper quality. We totally agree with the suggestion and wrote that in page 2:

When the voltage in M0 (VLS) changes from -15V to 15V, Vth shifts linearly from 8.5V to -4V, and the working mode of TFTs changes from the enhancement characterization to the depletion characterization.

Location of Change: Page 2-3, Line 97-99

  1. Reviewer2 wrote:

Lines 104-105. Please define the meanings of the cascade signal M(n) and Q point.

Our Response:

It should be pointed out that the G(n) is the output signal from the SR circuit and act the scan signal in AMOLED displays. The function of M(n) not only act the step-in signal from the next SR circuit, but the feedback signal for the previous SR circuit.

Location of Change: Page 4, Line 121-124

  1. Reviewer2 wrote:

 Is it the same working point of Qb in Figs. 4 (a) with the working point of QB in Fig. 4(b).

Our Response:

We greatly appreciate this question, which help us further improve paper quality.

Qb is the same working point of QB. We have changed from the description of QB to Qb in Fig.4b.

Fig.4b operation waveform of the proposed shift register circuit

  1. Reviewer2 wrote:

Lines 113-115. The second step (P2): VV1 is changed to VGL. T11 and T12 are turned off. At this time, VQ maintains VGH-VGL1-VGL2. Please check if VQ maintains VGH-Vth1-Vth2.

Our Response:

We greatly appreciate this question, which help us further improve paper quality. We totally agree with the suggestion and wrote that in Page5.

VV1 is changed to VGL. T11 and T12 are turned off. At this time, VQ maintains VGH-Vth1-Vth2.

Location of Change: Page 5, Line 134-135

  1. Reviewer2 wrote:

Please revise 60 HZ to 60 Hz in Table2

Our Response:

We greatly appreciate this question, which help us further improve paper quality. The frame rate has changed from 60 HZ to 60Hz

Location of Change: Page 9, Table 2

Round 2

Reviewer 1 Report

Please provide the information about the TFT device model used in the simulation studies.

Author Response

  1. Reviewer1 wrote:

Please provide the information about the TFT device model used in the simulation studies.

Our Response:

We greatly appreciate this question, which help us further improve paper quality. We totally agree with the suggestion and increase the recent advances in SR technology as follow:

The Smart Spice is used to make the accurate SPICE model. The initial model parameters are mainly determined by the measured TFT characteristics. The Fig.5a shows the transfer curves of TFT with device size of W/L=20mm/8mm. The model process can be described as follows: Firstly, we extract RPI model with parameter from the measured oxide TFT characteristics. Secondly, logical ternary condition is used to eliminate noise like characteristics. Finally, the deviation between the measured TFT characteristics and the simulated transfer curves should be set to <5%. It can be seen that the simulated transfer curves of IGZO-TFTs fit well with the experimental results of the fabricated TFTs in Fig.5a. Smart Spice (Silvaco) simulation is used to testify the feasibility of the SR circuits.

Location of Change: Page 5    line 143-151

Figure 5. (a) The experimental result and device model of the transfer characteristics of IGZO-TFTs